# Recent Advancement in Diagnosis of Biliary Tract Cancer through Pathological and Molecular Classifications

**DOI:** 10.3390/cancers16091761

**Published:** 2024-05-01

**Authors:** Sang-Hoon Lee, Si Young Song

**Affiliations:** 1Department of Internal Medicine, Konkuk University Medical Center, Konkuk University School of Medicine, Seoul 05030, Republic of Korea; lshjjang_2000@hanmail.net; 2Department of Internal Medicine, Yonsei University College of Medicine, Seoul 03772, Republic of Korea

**Keywords:** biliary tract cancer, cholangiocarcinoma, gallbladder cancer, diagnosis, precision medicine

## Abstract

**Simple Summary:**

The development of various molecular techniques has led to the introduction of a new classification for biliary tract cancer and a better understanding of the clinicopathological features of the disease. Furthermore, as new diagnostic modalities and research findings have been published, they enable accurate diagnoses, differentiations, and clinical assessments based on the characteristics of each subtype. This article reviews the current imaging and histologic diagnostic techniques along with future perspectives on molecular diagnosis, to approach precision medicine for biliary tract cancer.

**Abstract:**

Biliary tract cancers (BTCs), including intrahepatic, perihilar, and distal cholangiocarcinomas, as well as gallbladder cancer, are a diverse group of cancers that exhibit unique molecular characteristics in each of their anatomic and pathological subtypes. The pathological classification of BTCs compromises distinct growth patterns, including mass forming, periductal infiltrating, and intraductal growing types, which can be identified through gross examination. The small-duct and large-duct types of intrahepatic cholangiocarcinoma have been recently introduced into the WHO classification. The presentation of typical clinical symptoms, as well as the extensive utilization of radiological, endoscopic, and molecular diagnostic methods, is thoroughly detailed in the description. To overcome the limitations of traditional tissue acquisition methods, new diagnostic modalities are being explored. The treatment landscape is also rapidly evolving owing to the emergence of distinct subgroups with unique molecular alterations and corresponding targeted therapies. Furthermore, we emphasize the crucial aspects of diagnosing BTC in practical clinical settings.

## 1. Introduction

Biliary tract cancer (BTC) encompasses a range of invasive adenocarcinomas, including cholangiocarcinomas (arising in the intrahepatic, perihilar, or distal biliary tree), and gallbladder cancers. Cholangiocarcinomas arising from the bile ducts proximal to the second-order ducts are classified as intrahepatic cholangiocarcinoma (iCCA), those originating between the second-order ducts and the insertion of the cystic duct are termed perihilar cholangiocarcinoma (pCCA; previously referred to as Klatskin tumors), and those arising from the bile ducts distal to the insertion of the cystic duct are termed distal cholangiocarcinoma (dCCA). Extrahepatic cholangiocarcinoma collectively refers to pCCA and dCCA [1]. Gallbladder cancer (GBC) originates either from the gallbladder itself or from the cystic duct. 

BTCs exhibit heterogeneous clinical manifestations, molecular characteristics, and biological behaviors, depending on their anatomical, pathological, and molecular classifications. In recent years, increasing genomic research has begun to uncover the molecular underpinnings of BTC and offer many potential treatments, ushering in a new era in precision medicine. However, in addition to understanding the clinicopathologic development of each BTC subtype, there must be an individualized assessment of each subtype and an effort to overcome clinical diagnostic hurdles. Herein, we review the current techniques for the imaging and histologic diagnosis of BTC, along with future perspectives on molecular diagnosis, to approach precision medicine in BTC.

## 2. Pathologic Classification

The initial step in diagnosing BTC is to identify the anatomical location and growth patterns, followed by a microscopic assessment of the differentiation and subtype. These findings should be further supported by immunohistochemistry (IHC) staining, and, finally, molecular subtyping can be performed. Several classifications have been proposed based on the anatomical location, histopathological appearance, and molecular characteristics of BTC [1,2,3].

### 2.1. Pathologic Classification of Cholangiocarcinoma

iCCA is grossly classifiable into three growth patterns (Table 1): mass forming type (60–80% of cases) presents as a mass-like lesion within the hepatic parenchyma; periductal infiltrating type (15–35% of cases) is characterized by infiltration along the bile ducts and portal tracts, leading to strictures and thickening of the affected bile ducts, as well as the dilatation of the peripheral bile ducts; and intraductal growing type (6–29% of cases) consists of a polypoid or papillary tumor within the dilated bile ducts [1,4,5,6]. Macroscopically, pCCA and dCCA have similar growth patterns: the flat or nodular sclerosing type (73% of cases, corresponding to the features of periductal infiltrating type) and the intraductal papillary type (27% of cases) [7]. 

Conventional iCCA can be further divided into two main histologic types based on the size of the affected duct, which are small duct-type and large duct-type iCCA. These categories have been recently introduced into the WHO classification [8]. Small duct-type iCCA (36–84% of cases) is composed of the small-sized tubular growth of cuboidal or low-columnar tumor epithelial cells with little or no mucin production. Meanwhile, large duct-type iCCA (8–60% of cases) is characterized by mucin-producing columnar cells that form irregularly shaped and sized tubules or gland-like structures, which are commonly accompanied by an aggressive growth pattern and a desmoplastic reaction [8,9,10]. 

Cholangiocarcinoma can display various morphological subtypes, which may be attributed to different cell origins, carcinogenesis, and progression pathways. For instance, the canals of Hering and interlobular bile ducts are thought to be the cell origins for small duct-type iCCA, which typically presents as a mass-forming pattern in the context of chronic hepatitis and cirrhosis [11], whereas the peribiliary glands are likely the origin of large duct-type iCCA, pCCA, and dCCA, which lead to periductal-infiltrating lesions related to biliary inflammation, such as hepatolithiasis, parasite infection in the bile ducts, and primary sclerosing cholangitis (PSC) [7,11]. Notably, the intraductal growing pattern illustrates a separate pathway originating from the large bile ducts and is frequently linked to a more favorable prognosis.

In an investigation of IHC staining, both large and small duct-type iCCAs displayed positive staining for EMA (MUC1), HNF-1β, CK7, and CK19. The immunostaining for CK20 was typically negative or focally positive. Small duct-type iCCA demonstrated positive staining for NCAM (CD56), CRP, N-cadherin, and IDH 1/2, whereas large duct-type iCCA, similar to pCCA and dCCA, displayed positive staining for mucin (identified through mucicarmine or Alcian blue staining), MUC-5AC, MUC-6, S-100, TFF1, MMP, and KRAS [9,12]. 

The 2010 WHO classification suggested three types of precancerous lesions in the biliary tract: the flat type (biliary intraepithelial neoplasia, BilIN), the papillary type (intraductal papillary neoplasm of the bile duct, IPNB), and the cystic type (mucinous cystic neoplasm, MCN). Recently, intraductal tubular neoplasm of the bile duct (ITNB) was proposed as another candidate for preneoplastic lesions; however, its advanced form remains unclear [13]. IPNB may be associated with the intraductal growing type of iCCA and intraductal papillary type of pCCA and dCCA. BilIN may precede a periductal infiltrating (iCCA) and a flat or nodular sclerosing (pCCA and dCCA) pattern [14]. MCN can progress to an invasive carcinoma that may develop into a cystic lesion with a grossly surrounding nodular lesion. As of yet, no precursor lesions have been identified for mass-forming iCCA [7]. 

### 2.2. Pathologic Classification of Gallbladder Cancer

Adenocarcinoma is the main histological classification of GBC (approximately 90% of cases) [15]. GBC can exhibit an infiltrative, nodular, or papillary gross morphology or a combination of these morphologies. In addition, there are three premalignant lesions of gallbladder adenocarcinoma: adenoma, BilIN, and intracystic papillary neoplasm (ICPN). BilIN is invisible on gross inspection but can be microscopically identified around invasive tumors or chronic cholecystitis. ICPN is grossly identified as an exophytic polypoid mass or diffuse friable thickening of the mucosa and is composed of mucinous epithelial cells with papillary and tubular arrangements [16]. Dysplasia of the BilIN and ICPN is typically categorized using a three-tier system, with high-grade dysplasia placed in the same group as carcinoma in situ. The current definitions of adenoma and ICPN are unclear and require revised diagnostic criteria to ensure the consistency and accuracy of diagnoses.

## 3. Molecular Classification

In addition to the heterogeneity in anatomical locations and growth patterns, molecular profiling studies have revealed significant molecular heterogeneity across BTCs [17,18,19]. Driver mutations susceptible to targeted therapy have been identified in each BTC subtype, which are typically mutually exclusive from one another (Figure 1). For example, small duct-type iCCA is enriched for actionable targets, such as *IDH 1/2* mutations (15–20%) and *FGF2* fusions (10–20%). Large duct-type iCCA tumorigenesis frequently involves *KRAS* (15–30%) and *TP53* mutations (10–40%). GBC, pCCA, and dCCA are characterized by a high frequency of *KRAS* mutations (30–45%), *ERBB2* amplification (15–20%), and a low frequency of *IDH 1/2* or *FGFR2* fusions. Although rare, gene rearrangements, such as *NTRK*, *ROS1*, or *ALK* fusions, have been identified in BTC. All subtypes of BTC harbor similar rates of *BRAF* alteration (3–5%), homologous recombination deficiency (5–15%), and microsatellite instability-high (MSI-H)/mismatch repair (MMR)-deficiency (dMMR; 2–5%) [20,21,22]. 

Despite this heterogeneity, recurring molecular subtypes with driver mutations susceptible to targeted therapy have been found. These subtypes are typically mutually exclusive of one another. For example, fibroblast growth factor receptor (FGFR)-2 gene translocations and isocitrate dehydrogenase-1 (IDH1) mutations occur nearly exclusively in iCCA, while KRAS proto-oncogene (KRAS) mutations and receptor tyrosine-protein kinase erbB-2 (ERBB2) amplifications are more common in pCCA, dCCA, and GBC [19,23,24]. Targeted therapy, such as FGFR inhibitors [25,26,27], mutant-IDH inhibitors [28], BRAF inhibitors [29], and HER2 inhibitors [30,31,32], as well as immunotherapy used alone [33] or in combination [34,35], have undergone clinical studies and have recently been incorporated into clinical practice to treat patients with BTC. 

### 3.1. Molecular Classification of Cholangiocarcinoma

Integrated multi-omics analyses of cholangiocarcinoma have led to a deeper understanding of cancer traits, resulting in the proposal of several molecular classifications (Table 2). These molecular-subtyping methods are based on genomic, transcriptomic, proteomic, or tumor microenvironment (TME) analyses, and they primarily include samples of iCCA. While molecular classifications are evolving to become increasingly refined and treatment-matched, further studies are needed to better understand the pathological–molecular correlations and enter the era of precision medicine.

### 3.2. Molecular Classification of Gallbladder Cancer

Previous molecular studies have generally focused on characterizing the differences between cholangiocarcinomas rather than specifically examining GBC due to its rarity and relatively low mutation burden [21,41]. Nepal et al. recently investigated the GBC prognostic subtypes (subtype 1~3) based on their molecular profiles using an integrative multi-omics approach [42]. Subtype 2 is linked to a positive prognosis, characterized by higher levels of immune infiltrates and a gastric foveolar-like histomorphology. In contrast, subtypes 1 and 3, which have poor survival, were associated with an advanced stage, immunosuppressive TME features (myeloid-derived suppressor cell accumulation, extensive desmoplasia, and hypoxia), and T-cell dysfunction [42]. Other next-generation sequencing (NGS) data for GBC samples indicated that molecular alterations were distributed differently across GBC pathologic subtypes [43].

## 4. Clinical Presentation

The appearance and characteristics of clinical symptoms are contingent upon the anatomical site of the primary tumor and its associated metastasis. Patients with extrahepatic cholangiocarcinoma typically become symptomatic when biliary obstruction caused by the disease results in jaundice. Patients with iCCA are less likely to experience jaundice and instead exhibit non-specific symptoms, such as dull right upper quadrant pain or unexplained weight loss. Approximately 20–25% of patients are asymptomatic, with the lesions detected incidentally [44]. Patients with early GBC are also usually asymptomatic and are often diagnosed incidentally through preoperative imaging studies or intra- or postoperative examinations. 

Laboratory tests are generally not useful for diagnosis; high levels of alkaline phosphatase or serum bilirubin may suggest biliary obstruction. Serum tumor markers such as carcinoembryonic antigen (CEA) and carbohydrate antigen 19-9 (CA 19-9) are frequently elevated but do not provide diagnostically useful results due to their lack of specificity and sensitivity [45,46,47]. Although an established diagnosis is present, tumor markers can still offer useful information about the response to treatment and prognosis.

## 5. Diagnostic Tool

### 5.1. Ultrasonography

Many patients initially undergo transabdominal ultrasonography to assess the biliary tree, and the results may aid in identifying the location of the lesion: an abrupt change in the extrahepatic duct diameter with intrahepatic and extrahepatic biliary dilatation (dCCA case); intrahepatic ductal dilatation with normal-caliber extrahepatic ducts (pCCA case); mass lesions, occasionally in a non-cirrhotic liver, without radiographic characteristics of hepatocellular carcinoma (HCC) (iCCA case); or a protruding mass in the gallbladder, which sometimes extends directly into the liver bed (GBC case).

### 5.2. CT and MRI

Computed tomography (CT) is the standard method for diagnosing and staging BTC. It offers a thorough assessment of the primary tumor, taking into account its relationship with adjacent structures (specifically, portal vein and hepatic artery involvement, determining resectability), and potential thoracic and abdominal spread [48]. Magnetic resonance cholangiography (MRI) has comparable accuracy to CT for diagnosis and staging, but it includes particular sequences such as diffusion-weighted imaging (DWI) and the capability to carry out magnetic resonance cholangiopancreatography (MRCP), which is crucial for pCCA staging [49]. 

#### 5.2.1. Radiologic Findings of Mass-Forming Cholangiocarcinoma

The most common imaging pattern of mass-forming iCCA in both CT and MRI is characterized by an arterial peripheral rim enhancement that progresses centripetally with homogeneous contrast agent uptake, which continues until the delayed phase or remains stable during the late dynamic phases [49]. In the hepatobiliary phase of gadoxetic acid-enhanced MRI, most mass-forming iCCA exhibits ‘EOB-cloud’, which features a central area of mild hyperintensity that is cloud-like in appearance, surrounded by a hypointense area in the periphery of the tumor [50]. However, as no specific radiological pattern exists, histopathological or cytological results are necessary to confirm the diagnosis. 

The primary differential diagnoses for mass-forming iCCA are HCC, metastatic adenocarcinoma, inflammatory pseudo-tumors, and angiosarcoma. It is crucial to differentiate HCC, the most common primary hepatic malignancy, due to its different prognoses and treatments. Early arterial enhancement and washout of contrasts are the key patterns in favor of HCC, while capsular retraction and peripheral bile duct dilatation are more suggestive of iCCA. The target sign in DWI, defined as central hypointensity and a peripheral hyperintense rim, helps in the distinction of iCCA from HCC [51]. However, scirrhous HCC can be challenging to distinguish from mass-forming iCCA in imaging, making it necessary for tissue diagnosis. Metastatic adenocarcinoma can show many typical findings of iCCA, including central hypointensity or intrahepatic bile duct dilatation. It can also be difficult to differentiate based on histopathology and requires special immunohistochemical studies [52]. Therefore, when approaching a suspected mass-forming iCCA, it is essential to exclude extrahepatic primary malignancies, especially colorectal cancer.

#### 5.2.2. Radiologic Findings of Periductal-Infiltrating Cholangiocarcinoma

The most common growth pattern in pCCA and dCCA is characterized by a narrowed biliary duct displaying irregular circumferential wall thickening (usually with a thickness of ≥5 mm) along with upstream biliary tree dilatation. These tumors slowly enhance to a peak in the delayed phase; however, they are rarely hypervascular and are enhanced in the arterial phase [53]. When infiltration is nodular, the bile ducts appear protuberant, whereas they appear narrowed and stretched when infiltrated diffusely [54].

Periductal-infiltrating cholangiocarcinoma can be misleadingly similar to other biliary diseases such as PSC, Mirizzi syndrome, portal biliopathy, IgG4-related sclerosing cholangitis, benign idiopathic stricture, and hepatobiliary sarcoidosis or lymphoma. PSC typically presents as multiple intra- and extrahepatic biliary strictures with a beaded appearance in MRCP. The intervening segments exhibit only slight dilation, with the strictures typically being quite short in length [55]. On MRCP of Mirizzi syndrome, an abrupt stricture of the common hepatic duct is evident, along with a normal common bile duct and an impacted gallstone located in the neck of the distended gallbladder [56]. Portal biliopathy refers to the narrowing of the extrahepatic biliary tract as a result of extrahepatic portal vein obstruction. This leads to the obstructive effect of peribiliary collateral vessels or ischemic damage to the biliary tract. Imaging studies revealed a circumferential, long, and smooth stricture of the common bile duct accompanied by the presence of collaterals and choledochal varices [57]. IgG4-related sclerosing cholangitis is a chronic inflammatory condition affecting the biliary system, which is frequently observed in conjunction with other manifestations of IgG4-related disease. IgG4-related sclerosing cholangitis shows circumferential symmetric wall thickening of the bile ducts, frequently involving the extrahepatic segments, and features smooth outer and inner margins, a visible lumen in the thickened segments, and delayed homogenous contrast enhancement [58]. Nine out of ten cases exhibit pancreatic involvement, which typically presents with diffuse or focal pancreatic enlargement, a peripheral capsule-like rim, and a pancreatic duct stricture [59]. The diagnosis of IgG4-related sclerosing cholangitis is typically made by combining several factors, including characteristic imaging results, serum IgG4 antibody levels, histological findings, and the patient’s response to steroid therapy [60]. 

#### 5.2.3. Radiologic Findings of Intraductal-Growing Cholangiocarcinoma

The reported incidence of the intraductal-growing type ranges between 8% and 18% of all types of cholangiocarcinoma [49]. These tumors generally appear as polypoid or sessile masses that are restricted within the bile duct, accompanied by proximal ductal dilatation caused by either occlusion or the overproduction of mucin. These lesions exhibit imaging characteristics similar to those of mass-forming types, displaying a heterogeneous enhancement that begins early and reaches its peak in the delayed phase. A significantly dilated intrahepatic bile duct segment can give the appearance of a cystic mass such as cystadenoma, cystadenocarcinoma, or a liver abscess [61]. 

#### 5.2.4. Radiologic Findings of Gallbladder Cancer

GBC can present in various ways, such as a polypoid mass protruding into the lumen or completely filling it, focal or diffuse wall thickening, or a substantial mass in the gallbladder fossa with an indistinguishable gallbladder [62]. The indicators of GBC complicated by cholecystitis, rather than simple cholecystitis, include a higher frequency of lymph node enlargement, a more extensive wall thickness, focal irregularity in the wall thickness, and less distention of the gallbladder [63]. 

### 5.3. PET-CT

Positron emission tomography (PET)-CT can be employed to complement CT and MRI in order to provide additional information about lymph node involvement, the presence of distant metastasis, and postoperative recurrence. In fact, preoperative PET scanning has been shown to result in a change in surgical management in approximately one-fourth of cases, primarily by detecting occult distant metastases. However, due to its low specificity, it is not sufficient for the diagnosis of primary lesions, and cytological or histological confirmation is still necessary [64]. 

### 5.4. EUS

Endoscopic ultrasound (EUS) is a diagnostic tool that can visualize the local extent of the primary tumor and the status of the regional lymph nodes, particularly in cases where dCCA lesions are suspected. EUS-guided fine needle aspiration (FNA) of tumors and enlarged lymph nodes can also be performed. EUS-FNA has a higher sensitivity for detecting malignancies in distal tumors than endoscopic retrograde cholangiopancreatography (ERCP) with brushings [65]. However, EUS has been found to be less effective in imaging and staging proximal lesions compared with distal lesions, and clinical experience with this technique is relatively sparse [66].

EUS is also considered a useful modality for both detecting and distinguishing gallbladder polyps, as well as for staging early GBC. In particular, EUS is helpful in assessing the depth of tumor invasion within the gallbladder wall [67,68] and defining lymph node involvement in the portal hepatis or peripancreatic regions. Although some authors have reported accurate and safe results of EUS-FNA for GB wall lesions [69], this procedure poses a potential risk of bile leakage after gallbladder biopsy. When comparing the diagnostic performance and safety of EUS-FNA in patients with suspected GBC between GB samples and lymph node samples, EUS-FNA showed a safe and high diagnostic performance regardless of the target site. In particular, endoscopists preferred lymph node sampling in the following clinical situations: GB lesions of <4 cm in size, a wall-thickening type, a fundal location, and an absence of liver invasion [70]. 

### 5.5. ERCP or Percutaneous Transhepatic Cholangiography (PTC)

Preoperative cholangiography, which can be performed using either ERCP or PTC, may be necessary either for diagnostic or therapeutic purposes for patients with biliary obstruction. Recently, MRCP or CT scanning, which is non-invasive and highly accurate, has largely replaced invasive cholangiography for diagnostic purposes. 

#### 5.5.1. Intraductal Ultrasound (IDUS)

IDUS uses a small wire-guided ultrasound catheter that provides high-resolution images, enabling the precise evaluation of the biliary tract during ERCP. The utility of IDUS lies in its ability to characterize malignant biliary strictures and determine the local staging of cholangiocarcinoma. It can detect early lesions in the biliary tree, estimate the longitudinal tumor extent, and identify tumor infiltration into adjacent organs (e.g., the pancreas) and major vessels (e.g., the portal vein and hepatic artery) [71,72,73]. Unlike EUS, IDUS is frequently more effective in assessing the proximal biliary system and surrounding structures, including the right hepatic artery, portal vein, and hepatoduodenal ligament. However, IDUS limits the evaluation of more distant tissues or lymph nodes and cannot be used to perform FNA.

#### 5.5.2. Peroral Cholangioscopy (POC)

POC, which entails the direct visualization of the bile ducts using a highly specialized cholangioscope during ERCP, is a valuable tool for assessing indeterminate biliary strictures. For example, in the cases of biliary strictures where sampling techniques, such as brush cytology or biopsy, during routine ERCP are unable to determine whether they are benign or malignant, POC with targeted biopsies of bile duct lesions can provide a more accurate diagnosis for indeterminate strictures [74,75,76,77]. It can also be used to investigate equivocal ERCP findings, evaluate the extent of cholangiocarcinoma before surgery, and identify undetectable stones using conventional cholangiography. “Tumor vessels” may be observable during POC in patients with cholangiocarcinoma, which are characterized by irregularly dilated and tortuous blood vessels. Other characteristic findings suggesting malignancy include nodules or masses, infiltrative or ulcerative strictures, and papillary or villous mucosal projections [78]. A recent study reported 100% sensitivity and 89.5% specificity for visual impressions during POC examinations [79]. 

#### 5.5.3. Tissue Biopsy

ERCP or PTC-guided biopsies and brush cytology are the traditional standard methods for the tissue diagnosis of periductal-infiltrating or intraductal-growing cholangiocarcinoma. Brush cytology is a highly specific diagnostic tool; however, its low sensitivity is a significant drawback (e.g., 97% specificity and 43% sensitivity for detecting cholangiocarcinoma in patients with PSC) [80]. The incorporation of endoscopic biopsies for malignant strictures increases the diagnostic accuracy to only 43–88% [81,82,83]. These diagnostic tests can be beneficial if they yield positive results, but they cannot entirely exclude the possibility of malignancy if their results are negative.

Fluorescence in situ hybridization (FISH) is a cytological technique that makes use of labeled DNA probes to detect any abnormal loss or gain of chromosomes or chromosomal loci in cells routinely collected through the brush technique. This method can improve the sensitivity of brush cytology [84]. A meta-analysis of FISH demonstrated that this method is highly specific, with a pooled specificity of 70%, but it has limited sensitivity (68%) for the diagnosis of cholangiocarcinoma in patients with PSC [85]. 

Another auxiliary technique for improving the diagnostic ability of bile cytology is the implementation of a new scoring system for evaluating cytologic results. Hayakawa et al. introduced a scoring system based on four cytological features: abnormal chromatin, irregularly arranged nuclei, irregularly overlapped nuclei, and irregular cluster margins. The scoring system yielded an area under the receiver operating characteristic (ROC) curve (AUC) of 0.981, with a sensitivity of 87% and specificity of 98% [86]. A different study reported that the diagnostic sensitivity of bile cytology increased from 31.6% to 80.3% after combined p53 immunostaining [87].

### 5.6. Liquid Biopsy Based on Bile Samples

Liquid biopsy is a blood test that identifies circulating tumor cells, cell-free nucleic acids, and secreted proteins present in body fluids, such as blood, urine, saliva, and bile. Unlike tissue biopsy, liquid biopsy is less invasive, and it is easier to obtain biological fluids. Due to the distinct anatomical location of BTC, bile is recognized as a promising body fluid for diagnosing BTCs. 

Recently, various emerging analytical methods for extracellular vesicles (EVs), nucleic acids, proteins, and metabolites in bile have been developed as potential biomarkers for BTC diagnosis [88]. For example, circular RNA (Circ-CCAC1) in serum-derived or bile-derived EVs has a diagnostic role, with an AUC of 0.857 [89]. In a prospective study on bile samples, *KRAS* mutations detected in bile cell-free DNA indicated the possibility of cholangiocarcinoma in high-risk lesions such as PSC [90]. A study screened four DNA methylation biomarkers (*COD1*, *CNRIP1*, *SEPT9*, and *VIM*) based on DNA methylation analysis of ERCP brush samples and achieved 85% sensitivity and 98% specificity with an AUC of 0.944 [91]. Based on this study, the role of a four-gene methylation panel in bile was investigated to predict early diagnosis of BTC in patients with PSC. The findings indicated that the AUC for predicting the diagnosis of cholangiocarcinoma in patients with PSC within one year ranged from 0.84 to 0.98, with a sensitivity of 67–96% and a specificity of 93–98% [92]. sB7-H3, a cancer-related immune protein, is elevated in the bile of patients diagnosed with malignant biliary obstruction, including BTC and pancreatic cancer. The ROC-AUC for diagnosing malignant biliary obstruction was 0.878, with a sensitivity and specificity of 81.2% and 81.6%, respectively [93]. A different study uncovered the usefulness of bile multi-omics analysis that incorporates metabolomics for the molecular diagnosis of GBC by integrating lipidomics and metagenomics in bile to define microbial and lipid variations that contribute to the onset of GBC. Using the random forest classifier model, this research developed a diagnostic model comprising eight lipid substances that can accurately distinguish GBC from gallstones or healthy groups, with an AUC of one [94]. Table 3 summarizes the previous studies on bile EVs, nucleic acids, and protein detection for the diagnosis of BTC (Table 3). Despite the advances in the use of liquid biopsy in bile for BTC diagnosis, more research is still needed to improve its sensitivity and specificity and validate it in large sample studies before its translation to routine clinical practice.

### 5.7. Liquid Biopsy Based on Blood Samples

Blood-based liquid biopsy has numerous potential applications in managing BTC. These include aiding in diagnosis as an adjunctive method when traditional investigations are deemed unfeasible or inconclusive, conducting risk stratification and prognosis evaluations, developing personalized medicine strategies, and identifying relapse and emerging resistance mechanisms [138]. 

Cell-free DNA (cfDNA) and circulating tumor DNA (ctDNA) are considered to reflect changes in tumor aggressiveness and size as they have been detected in both the tumor tissues and blood samples of patients with BTC [139]. According to Kumari et al., higher levels of cfDNA can assist in distinguishing between benign conditions and GBC, and they are associated with the burden of the tumor [140]. Another study reported that the methylation levels of *OPCML* and *HOXD9*, as determined in cfDNA, can help discriminate between cholangiocarcinoma and benign biliary disease [141]. In addition, several blood-based cfDNA and ctDNA assays feature a significant level of concordance with tumor tissue analyses [139,142,143,144]. Moreover, Okamura et al. indicated a higher concordance when comparing matching ctDNA-metastatic tumor tissues with ctDNA-primary tumor tissues [145]. Blood-based liquid biopsy can also be used to identify clinically relevant genomic alterations when guiding precision medicine [144,146,147,148]. A retrospective analysis of the ClarIDHy trial, which examined the efficacy of the IDH1 inhibitor ivosidenib in mutant *IDH1* cholangiocarcinoma, showed that the clearance of *IDH1* mutation, as assessed in plasma ctDNA in patients monitored over time, was associated with disease control [149]. De novo multiple point mutations within the FGFR2 kinase domain have been identified in post-progression cfDNA samples from patients with cholangiocarcinoma who developed resistance to FGFR inhibitors [150]. Yang et al. reported that lower copy number variations (CNVs) detected in cfDNA could predict favorable responses to immunotherapy in 187 hepatobiliary cancer patients [151].

Proteins and cytokines serve as potential diagnostic and prognostic biomarkers for liquid biopsies. CYFRA 21-1 [152,153], MMP-7 [154,155], osteopontin [156], periostin [157], and IL-6 [158] were found to be increased in the sera of patients with cholangiocarcinoma compared with healthy individuals and patients with benign biliary disease, including PSC. Among these diagnostic biomarkers, elevated CYFRA 21-1 and osteopontin levels demonstrated superior diagnostic potential for cholangiocarcinoma compared with CA 19-9 and CEA [152,158]. In addition, increased periostin levels have been identified as an independent predictor of overall survival in patients with iCCA [157]. The diagnostic utility of proteomic signatures in serum EVs for patients with iCCA, HCC, and PSC has been found to be effective, with a high degree of accuracy for the differential diagnosis of these liver diseases. These proteomic signatures have demonstrated higher AUC values than both CA 19-9 and α-fetoprotein levels [159]. Another study proposed an algorithm containing six serum metabolites that could differentiate iCCA from HCC or PSC (AUC: 0.9) [160]. 

## 6. Clinical Aspects for Pathologic and Molecular Diagnosis

### 6.1. Pathologic Diagnosis

Pathologic diagnosis for patients suspected of having BTC can be determined using a range of methods, including ERCP or PTC-guided biopsy, brush cytology, EUS-FNA, and ultrasonography/CT/MRI-guided biopsy. However, obtaining tissue can be challenging, especially in patients with perihilar lesions. In cases of potentially resectable tumors with typical findings of malignant biliary obstruction, a solitary intrahepatic mass, or early GBC confined to the gallbladder, surgery can be performed without preoperative pathologic diagnosis. In patients with biliary obstruction resulting from pCCA and dCCA without extraductal metastasis, it is recommended to perform ERCP or PTC-guided biopsies or brush cytology to obtain sufficient tissue for pathological diagnosis and molecular profiling. EUS-FNA is a potential alternative approach for obtaining biopsies of enlarged regional lymph nodes or distally located tumors. It may be considered if ERCP or PTC-guided biopsies yield negative or inconclusive results. In addition, EUS-FNA and ultrasonography/CT/MRI-guided biopsy via the transperitoneal approach rarely result in the seeding of tumor cells in the biopsy tract [161]. Therefore, it is necessary to establish tissue diagnosis prior to surgery in a multidisciplinary setting. 

Pathological diagnosis is important in the following situations: clinically indeterminate strictures, patients requiring diagnostic documentation before nonsurgical treatment, and situations where a physician or patient is hesitant to proceed with surgery without tissue diagnosis [162]. Conversely, tissue diagnosis is not mandatory for unresectable patients who are scheduled to receive only palliative management, such as biliary drainage. 

### 6.2. Molecular Diagnosis

Molecular profiling is recommended for advanced diseases and is considered suitable for systemic treatments [163]. Parallel tests for several genes using focused NGS are preferred over single-gene sequencing. NGS can be performed on formalin-fixed and paraffin-embedded tumor tissues, making it an excellent option for tissue biopsies. In cases where adequate tumor tissue is not available for NGS, liquid biopsies that utilize cell-free circulating DNA can be considered as an alternative [163]. The MSI status can be evaluated by IHC staining for MMR proteins, including MLH1, MSH2, MSH6, and PMS2. DNA-based assays can be used to analyze the composition and length of microsatellites. The preferred methods for NGS, IHC staining, or RNA sequencing depend on the target of interest and availability of materials, such as tissue or ctDNA.

## 7. Approach to the Patient

### 7.1. Suspected iCCA

When an intrahepatic lesion is suspected, cross-sectional imaging (multiphasic contrast-enhanced CT or MRI) is performed to differentiate between HCC and mass-forming iCCA. However, the classical radiologic features of iCCA are present in only 70% of cases [50], and some small mass-forming iCCAs may resemble HCC, displaying hyperenhancement during the arterial phase and washout during the delayed phase. If the initial imaging test is non-diagnostic, other imaging modalities (CT or MRI) can be conducted. A biopsy or surgery of the lesion is performed if the diagnosis remains uncertain. 

The complexity of the issue is exacerbated by the presence of mixed hepatocellular–cholangiocellular carcinomas, where both cholangiocarcinoma and HCC elements are found in the same nodule [164]. Studies have suggested that these tumors exhibit a unique appearance in cross-sectional imaging examinations. A mixed hepatocellular–cholangiocellular carcinoma is indicated by a strongly enhanced rim and an irregular shape in gadoxetic acid-enhanced MRI, while a mass-forming iCCA is suggested by a lobulated shape, weak rim, and target appearance [165]. The target appearance can also be used to distinguish mixed hepatocellular–cholangiocellular carcinomas from atypical hypovascular HCC [166]. Additionally, the existence of liver capsule retraction and biliary dilatation near an intrahepatic lesion may lead to the suspicion of an iCCA diagnosis; however, a biopsy may be needed to confirm the diagnosis. These mixed tumors are staged as iCCA and not as HCC.

IHC staining of tissue biopsies is required to differentiate iCCA from metastatic lesions and mixed hepatocellular–cholangiocellular carcinoma. Tumors that test negative for TTF-1 (lung), CDX2 (colon), and DPC4 (pancreas) and positive for AE1/AE3, CK7, and CK20 (biliary epithelium) are indicative of iCCA [167]. 

### 7.2. Suspected pCCA

Undertaking a thorough assessment with cross-sectional imaging studies (particularly, enhanced MRI with MRCP is preferred over CT) and EUS is crucial for defining the tumor location, size, morphology, involvement of the hepatic artery or portal vein, volume of the potential liver remnant, lymph node involvement, and presence of distant metastases. If imaging studies and/or tissue samples strongly indicate pCCA, the tumor staging proceeds directly. In situations where the diagnosis is uncertain, we generally opt for an ERCP procedure that incorporates brush cytology (with or without IDUS). When feasible, performing POC to evaluate the bile ducts can be considered. Alternatively, an MRI- or CT-guided biopsy can be carried out if the imaging reveals a mass lesion, although there is a small risk of needle tract seeding. In cases where the diagnosis remains uncertain, surgical intervention may be necessary to establish a diagnostic confirmation.

### 7.3. Suspected dCCA

In cross-sectional imaging, dCCA may be observed as an abrupt narrowing of the bile duct accompanied by upstream biliary dilatation. Typically, a nodular mass or concentric and asymmetric thickening of the bile duct with enhancement often occur together. In uncommon cases of thickening or stricture of the distal bile ducts without the presence of a mass, it is difficult to differentiate them from benign strictures. While ERCP has traditionally dominated the initial workup of dCCA owing to tissue sampling for diagnosis and biliary decompression, EUS has recently become the preferred method for the direct visualization and sampling of the distal bile duct. ERCP carries a risk of enhancing cholangitis by injecting contrast, whereas EUS-FNA poses a risk of seeding the biopsy tract. If the radiographic findings for dCCA are conclusive enough that a negative biopsy would be considered a potential false-negative and the tumor appears resectable, then a biopsy is unnecessary.

### 7.4. Patients with PSC

PSC is a prevalent risk factor for BTC, with the incidence of cholangiocarcinoma in PSC patients estimated to be 5–10% [168,169]. Cholangiocarcinoma in PSC usually infiltrates and manifests as progressive strictures in the perihilar areas [169,170]. In these cases, patients may sometimes have a dominant benign biliary stricture that is difficult to differentiate from cholangiocarcinoma. Rarely do mass lesions appear in imaging scans, and patients typically do not exhibit substantial intrahepatic biliary dilation. However, the discovery of a new parenchymal lesion near the bile ducts, the sudden emergence of bile duct dilatation, and the presence of disproportionate regional/segmental bile duct dilatation are indicators of potential cholangiocarcinoma development in these patients. 

In patients with PSC and suspected cholangiocarcinoma, CA 19-9 levels greater than 129 U/mL were found to be 79% sensitive and 98% specific in confirming the diagnosis [171]. However, the positive predictive value for cholangiocarcinoma, which is the likelihood of a patient with PSC and a CA 19-9 level of ≥129 unit/mL having the disease, was only 57%.

The initial step of the work-up is MRCP, which helps to determine the segmental extent of ductal involvement, search for intrahepatic metastases, and identify any abnormalities in the ductal anatomy. If the MRCP is non-diagnostic or if a dominant stricture is detected, an ERCP or PTC is obtained with brush cytology, which is almost 100% specific; however, despite the use of FISH, only 40–70% of patients with PSC and cholangiocarcinoma can be properly diagnosed [172,173]. In cases of negative cytologic results, it is recommended to perform MRI/MRCP and/or ERCP plus CA 19-9 tests again within 3–6 months. 

## 8. Conclusions

BTC is a heterogeneous disease that arises from the biliary tree. Although it has historically been classified as a single disease, extensive molecular characterization has recently led to more informative anatomical, pathological, and molecular classification of BTC. The development of radiologic and endoscopic tools for accurate diagnosis strengthens our understanding of BTC carcinogenesis. Precision medicine for BTC patients is facilitated by pathological and molecular profiling. We anticipate that advancements in diagnostic and personalized strategies for BTC management will lead to improved patient outcomes in the near future.

## Figures and Tables

**Figure 1 cancers-16-01761-f001:**
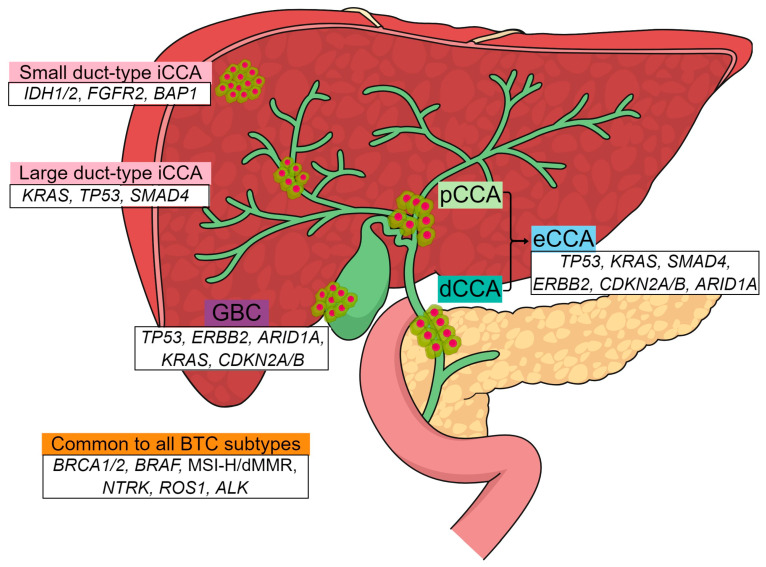
Molecular characteristics of biliary tract cancer according to anatomical location. iCCA, intrahepatic cholangiocarcinoma; pCCA, perihilar cholangiocarcinoma; eCCA, extrahepatic cholangiocarcinoma; dCCA, distal cholangiocarcinoma; GBC, gallbladder cancer.

**Table 1 cancers-16-01761-t001:** Clinicopathological features of cholangiocarcinoma.

Cholangiocarcinoma Type	Growth Pattern	Precancerous Lesion	Main Etiology
iCCA—small-duct type	Mass forming	None	Chronic hepatitis Cirrhosis
iCCA—large-duct type	Periductal infiltrating	BilIN	Hepatolithiasis Liver flukes PSC
	Intraductal growing	IPNB, MCN, and ITNB	
pCCA—dCCA	Flat or nodular sclerosing	BilIN	
	Intraductal papillary	IPNB, MCN, and ITNB	

iCCA, intrahepatic cholangiocarcinoma; BilIN, biliary intraepithelial neoplasia; IPNB, intraductal papillary neoplasm of the bile duct; MCN, mucinous cystic neoplasm; ITNB, intraductal tubular neoplasm of the bile duct; pCCA, perihilar cholangiocarcinoma; dCCA, distal cholangiocarcinoma; PSC, primary sclerosing cholangitis.

**Table 2 cancers-16-01761-t002:** Molecular classifications of intrahepatic cholangiocarcinoma.

Reference	Tumor Type	*n*	Classification	Molecular Characteristics and Prognosis
Sia et al. [17]	iCCA	149	Inflammation class	Activation of inflammatory signaling pathways Overexpression of *IL-4* and *IL-10* (Th2 marker) Favorable prognosis
			Proliferation class	Activation of oncogenic signaling pathways Overexpression of *EGF*, *RAS*, *AKT*, and *MET* Worse prognosis
Andersen et al. [18]	Cholangiocarcinoma	104	Cluster 1	No *KRAS* mutation Absence or weak expression of *HER2* and *MET* Good prognosis
			Cluster 2	Enriched *VEGF/ERBB*, *CTNNB1/MYC*, and *KRAS* mutations Poor prognosis
Farshidfar et al. [36]	Cholangiocarcinoma	32	*IDH*-mutant cluster *	*IDH1/2* mutation Elevated mitochondrial gene expression Loss of function of *ARID1A* and *PBRM1*
			*CCND1* amplification cluster *	Highly hypermethylated
			*BAP1/FGFR* cluster *	*BAP1* mutation or *FGFR2* fusion
Jusakul et al. [19]	Cholangiocarcinoma	69	Cluster 1	*ARID1A*, *BRCA1/2*, and *TP53* mutations *ERBB2* amplification CpG island hypermethylation
			Cluster 2	Enriched in *TP53* mutations High expressions of *CTNNB1*, *WNT5B* and *AKT1*
			Cluster 3	High CNA burden Enriched immune-related pathways
			Cluster 4	*BAP1* or *IDH1/2* mutation High expression of *FGFR* family proteins CpG shore hypermethylation Favorable prognosis
Job et al. [37]	iCCA	78	Immune desert subtype	Minimal expression of all TME signatures
			Immunogenic subtype	High adaptive immune cell presence Strong activation of fibroblasts and inflammatory and immune checkpoint pathways Best prognosis
			Myeloid-rich subtype	Strong monocyte-derived myeloid cell signatures Weak lymphoid signatures
			Mesenchymal subtype	Strong activation of fibroblast signatures Worst prognosis
Dong et al. [38]	iCCA	262	S1	Enriched *KRAS* mutations Upregulated inflammatory pathways and immunosuppressive TME signature Worst prognosis
			S2	High expression of proteins related to CAFs and ECM (FAP, POSTN, and FLT1)
			S3	Enriched in *TP53* mutations Upregulated pathways of cell cycle and MAPK signaling
			S4	*FGFR2* alterations, and *BAP1* and *IDH1/2* mutations High expression of adhesion and biliary-specific proteins (ANXA4, KRT18, and EPCAM) Best prognosis
Martin-Serrano et al. [39]	iCCA	122	Immune classical	High infiltration of immune cells (type-1 IFN) Enriched in *TP53* mutations alone Elevated metabolic-related pathways
			Inflammatory stroma **	Abundance of stromal deposition, TGFβ signaling, and T cell exhaustion Enriched *KRAS* mutations alone
			Hepatic stem-like	High M2-like macrophage levels in TME *FGFR2* alterations, and *BAP1* and *IDH1/2* mutations Elevated stemness-related pathways (NOTCH and YAP1)
			Tumor classical **	Enriched in *TP53* mutations alone and co-occurrence of *TP53* and *KRAS* mutations High expression of cholangiocyte markers
			Desert-like	Scarce immune infiltration and abundance of Tregs in TME Enriched in *TP53* mutations alone Enriched in mitotic spindles and WNT/β-catenin signaling
Cho et al. [40]	iCCA	102	Metabolism	*IDH1* and *BAP1* mutations Favorable prognosis
			Stem-like	High expression of ALDH1A1 and ALDH families
			Poorly immunogenic	*TP53* and *KRAS* mutations Poor prognosis

iCCA, intrahepatic cholangiocarcinoma; CNA, copy number aberration; TME, tumor microenvironment; CAFs, cancer-associated fibroblasts; ECM, extracellular matrix; Tregs, regulatory T cells; ALDH, aldehyde dehydrogenase. * Survival is not significantly different between the clusters. ** Inflammatory stroma and tumor classical classes are linked to more aggressive disease, although they are not independent predictors of survival.

**Table 3 cancers-16-01761-t003:** Diagnostic studies of bile EVs, nucleic acids, and proteins.

Biomarkers	*n*	ROC-AUC	Sensitivity (%)	Specificity (%)	Reference
Exosomal cargoes					
MicroRNA (miR-191, miR-486-3p, miR-1274b, miR-16, and miR-484)	96		0.67	0.69	[95]
MicroRNA (miR-483-5p, and miR-126-3p)	92	0.81, 0.74	0.811, 0.73	0.811, 0.865	[96]
MicroRNA (miR-141-3p, miR-200a-3p, miR-200c-3p, miR-200b-3p, and ENST00000588480.1)	100	0.757~0.869	0.63~0.83	0.6~0.867	[97]
LncRNA (ENST00000588480.1 and ENST00000517758.1)	91	0.709	0.829	0.589	[98]
Circle-RNA (circ-CCAC1)	316	0.857			[89]
Protein (claudin-3/CLDN3)	20	0.945	0.875	0.875	[99]
DNA	20	0.667	0.33	1	[100]
*KRAS* mutation	115		0.25	0.96	[101]
*KRAS* mutation	46	0.738	0.476	1	[102]
*KRAS* mutation	43	0.742	0.526	0.958	[103]
*KRAS* mutation and TP53 mutation	109	0.564/0.508	0.279/0.047	0.848/0.970	[104]
*KRAS* mutation and TP53 mutation	50	0.783, 0.750	0.567, 0.5	1, 1	[105]
*KRAS* mutation and TP53 mutation	49	0.733	0.467	1	[106]
*TP53*, *ERBB2*, and *KRAS*	42	0.955	0.909	1	[102]
*KRAS*, *TP53*, *CDKN2A*, *SMAD4*, and *BRAF*	60	0.737/0.715	0.536/0.462	0.937/0.969	[107]
Promotor methylation *INK4a/ARF*	243	0.84~0.98	0.67~0.96	0.93~0.98	[92]
Promotor methylation of *COD1*, *CNRIP1*, *SEPT9*, and *VIM*	80	0.775	0.773	0.778	[108]
Methylation of *DKK3*, *p16*, *SFRP2*, *DKK2*, *NPTX2*, and *ppENK*	125		0.71~0.83	0.94	[109]
*CCND2*, *CDH13*, *GRIN2B*, *RUNX3*, and *TWIST1*	241		0.92	0.98	[110]
Gene mutations in *KRAS*, *TP53*, *SMAD4*, and *CDNK2A*; methylation changes in *SOX17*, *3-OST-2*, *NXPH1*, *SEPT9*, and *TERT*					
150 tumor-related genes (widely targeted deep sequencing)	10		0.947	0.999	[111]
520 tumor-related genes (widely targeted deep sequencing)	28		0.955		[112]
RNA					
Human telomerase reverse transcriptase mRNA	20		0.833	1	[113]
miR-9, miR-145, and miR-944	18	0.765~0.975			[114]
RNU2-1f	23	0.856	0.67	0.91	[115]
miR-412, miR640, miR-1537, and miR-3189	83	0.78~0.81	0.5~0.67	0.89~0.92	[116]
miR-30d-5 and miR-92a-3p	106	0.730, 0.652	0.811, 0.657	0.605, 0.667	[117]
Protein					
CEACAM6	73	0.74	87.5	69.1	[118]
CEACAM6	41	0.92	83.3	93.1	[119]
SVV and CA199	102	0.78, 0.75	67.3, 96.4	80.9, 46.7	[120]
MUC1	68	0.85	90.0	76.3	[121]
MUC4	134		27	93	[122]
MUC5AC	46	0.85	75	76.9	[123]
Mac-2BP	78	0.70	69	67	[124]
VEGF	53	0.89	99.3	88.9	[125]
MCM2 and MCM5	42	0.80			[126]
HSP27 and HSP70	20	0.86, 0.81	90, 80	90, 80	[127]
SSP411	67	0.913 *	90.0	83.3	[128]
NGAL	40	0.74	77.3	77.2	[129]
NGAL	38	0.76	94	55	[130]
LCN2/NGAL	144	0.81	87	75	[131]
S100P	24	0.861	92.9	70	[132]
sB7-H3	323	0.878	81.2	81.6	[93]
α-1-antitrypsin	8	0.833	80	75	[133]
Amylase	239	0.751	66	74	[134]
PE-3B/amylase	68	0.877	81.8	89.3	[135]
M2-PK	167		90.3	84.3	[136]
GSH, hydrogen peroxide, GPx, Fe^2+^, and FNTA	46	0.683~0.852	67.9~100	52.9~76.5	[137]

* Serum samples for ROC analysis.

## Data Availability

Data sharing is no applicable.

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
