# Peer review of "Recent Advancement in Diagnosis of Biliary Tract Cancer through Pathological and Molecular Classifications"

_cancers, 2024, doi:10.3390/cancers16091761_

Round 1

Reviewer 1 Report

Comments and Suggestions for Authors

This review summarized recent advances on Biliary Tract Cancer diagnosis using molecular and pathological classifications. A main limitation of the current version is the overlook of surveying quality statistical analysis on BTC studies. Given the heterogeneity nature of BTC, it is not clear whether or how the efforts from published studies have been devoted to conduct robust analysis that leads to findings with better reproducibility and reliability. Please refer to the review [1] below and provide a discussion. In particular, the authors have only shown summary statistics in pathological and molecular diagnosis of BCT and approach to patients without mentioning measures on statistical significance including p-values and confidence intervals. Are there any rigorous statistical analysis available to facilitate the diagnosis of BTC?

In addition, this review ignored public databases for BCT data including TCGA where multiple types of molecular measurements are available and demands tailored integration analysis [2]. How will the multi-omics integration studies elucidate the genetic and genomic basis of BCT, and influences the diagnosis of BTC? Furthermore,  as an important clinic endpoint, survival outcomes and relevant analysis on BCT should be included in the review.

References:

1.Wu, C., & Ma, S. (2015). A selective review of robust variable selection with applications in bioinformatics. Briefings in bioinformatics, 16(5), 873-883.

 2.Wu, C., Zhou, F., Ren, J., Li, X., Jiang, Y., & Ma, S. (2019). A selective review of multi-level omics data integration using variable selection. High-throughput, 8(1), 4.

Comments on the Quality of English Language

Minor editing of English language required

Author Response

<Reviewer 1>

This review summarized recent advances on Biliary Tract Cancer diagnosis using molecular and pathological classifications. A main limitation of the current version is the overlook of surveying quality statistical analysis on BTC studies. Given the heterogeneity nature of BTC, it is not clear whether or how the efforts from published studies have been devoted to conduct robust analysis that leads to findings with better reproducibility and reliability. Please refer to the review [1] below and provide a discussion.

References: 1.Wu, C., & Ma, S. (2015). A selective review of robust variable selection with applications in bioinformatics. Briefings in bioinformatics, 16(5), 873-883.

Answer: Thank you for your thoughtful comment regarding our article. We appreciate your feedback, and would like to address the concerns raised regarding the structured review. First, we would like to clarify that our article is a review rather that a systemic review, and we did not intend to rigorously address precise statistical methodologies for reproducibility and reliability. Instead, we provided an overview of main research findings related to the diagnosis of biliary tract cancers, primarily simple biomarker studies rather than bioinformatic studies. As you mentioned, these biomarker studies have small sample sizes and experimental methods, which limits their reproducibility and reliability. In addition, delving into bioinformatic methods of robust analysis, which you mentioned as reference, is beyond my expertise and I do not believe it is essential for clinicians involved in the diagnosis and treatment of biliary tract cancer patients. In my opinion, detailed statistical methodologies for molecular research may not be necessary for their clinical practice. Again, thank you for your time and consideration.

In particular, the authors have only shown summary statistics in pathological and molecular diagnosis of BCT and approach to patients without mentioning measures on statistical significance including p-values and confidence intervals. Are there any rigorous statistical analysis available to facilitate the diagnosis of BTC?

Answer: Thank you for your comment. For pathologic diagnosis, as it is intrinsic to the nature of pathologic studies, it cannot provide p-value or confidence intervals aside from certain proportions. Regarding molecular classification, we have expanded Section 3 (titled as 3. Molecular classification) to provide a comprehensive overview of the proposed molecular classifications for each type of biliary tract cancers. Furthermore, regarding the question "Are there any rigorous statistical analyses available to facilitate the diagnosis of BTC?", I didn't quite grasp its precise meaning. Diagnosis of biliary tract cancer is confirmed pathologically, and statistical analysis is not necessary for diagnostic certainty.

In addition, this review ignored public databases for BCT data including TCGA where multiple types of molecular measurements are available and demands tailored integration analysis [2]. How will the multi-omics integration studies elucidate the genetic and genomic basis of BCT, and influences the diagnosis of BTC? Furthermore, as an important clinic endpoint, survival outcomes and relevant analysis on BCT should be included in the review.

References: 2.Wu, C., Zhou, F., Ren, J., Li, X., Jiang, Y., & Ma, S. (2019). A selective review of multi-level omics data integration using variable selection. High-throughput, 8(1), 4.

Answer: Thank you for your insightful comment. As you mentioned, we have addressed your suggestion regarding the multi-omics integration studies of TCGA data and other public databases for biliary tract cancer through additional content in Section 3 (titled as 3. Molecular classification. Our aim was to strengthen the paper with comprehensive coverage of molecular classification, molecular characteristics, and its prognosis. In addition, we expanded our coverage of liquid biopsy based on blood samples (as section 5.7.). We sincerely hope that our efforts have contributed to enhancing the quality of the paper. Once again, we are grateful for your assistance in this process.

Reviewer 2 Report

Comments and Suggestions for Authors

This review presents cytological, histological and molecular diagnoses in addition to the imaging studies to date for biliary tract cancer.

This paper is well written and problem-free, but there are a few suggestions.

Comments

Lines 122 to 128, to the section on Ultrasonography’ in Chapter 4.

Lines 128 and below should be deleted.

Lines 259 to 303, to the section on ‘Cytological diagnosis’ in Chapter 5.

Section 6.2.1 is better after the ‘Suspected dCCA’ section.

Author Response

<Reviewer 2>

This review presents cytological, histological and molecular diagnoses in addition to the imaging studies to date for biliary tract cancer.

 This paper is well written and problem-free, but there are a few suggestions.

Comments

Lines 122 to 128, to the section on ‘Ultrasonography’ in Chapter 4.

Lines 128 and below should be deleted.

Answer: Thank you for your valuable feedback. We've made the changes you mentioned.

Lines 259 to 303, to the section on ‘Cytological diagnosis’ in Chapter 5.

Answer: Thank you for your valuable suggestion. We would like to share our perspective on the points you have raised. We believe that the mentioned content is not specific to realm of pathological cytologic diagnosis, but more importantly to endoscopic findings directly observed through peroral cholangioscopy (POC). Rather than moving this content to the cytologic diagnostics section, we felt it would be more beneficial to keep it in its current context. We look forward to your understanding on this matter.

Section 6.2.1 is better after the ‘Suspected dCCA’ section.

Answer: Thank you for your insightful feedback. We've made the changes you mentioned.

Reviewer 3 Report

Comments and Suggestions for Authors

The manuscript demonstrates a commendable level of proficiency in its articulate composition and logical presentation. While its readability is good, offering comprehensive coverage of various facets of cholangiocarcinoma, from clinical aspects to molecular pathology, it is crucial to address a few typos for further refinement. Additionally, there is a notable concern regarding significant overlap with previous works that needs attention. In light of these issues, which require a high level of commitment from the authors, I strongly recommend the incorporation of iconography in the form of figures. Specifically, at least one figure should schematically depict the diverse locations of the biliary pathways, accompanied by annotations highlighting the most prevalent molecular alterations. This visual enhancement will not only strengthen the manuscript's overall quality but also serve as a valuable aid for readers in comprehending the intricate details of cholangiocarcinoma.

Comments on the Quality of English Language

The quality of English language is good.

Author Response

<Reviewer 3>

The manuscript demonstrates a commendable level of proficiency in its articulate composition and logical presentation. While its readability is good, offering comprehensive coverage of various facets of cholangiocarcinoma, from clinical aspects to molecular pathology, it is crucial to address a few typos for further refinement. Additionally, there is a notable concern regarding significant overlap with previous works that needs attention. In light of these issues, which require a high level of commitment from the authors, I strongly recommend the incorporation of iconography in the form of figures. Specifically, at least one figure should schematically depict the diverse locations of the biliary pathways, accompanied by annotations highlighting the most prevalent molecular alterations. This visual enhancement will not only strengthen the manuscript's overall quality but also serve as a valuable aid for readers in comprehending the intricate details of cholangiocarcinoma.

Answer: We greatly appreciate your keen eye for detail and valuable suggestions regarding typos, redundancies, and content enhancements. We have meticulously refined the manuscript to address these issues comprehensively. Additionally, in accordance with your recommendation, we have created Figure 1. Thank you once again for your constructive feedback.

Round 2

Reviewer 1 Report

Comments and Suggestions for Authors

NA

Comments on the Quality of English Language

The authors declined to revise based on major comments. This review did not incorporate any statistical validation or discussion, and the uncertainty in BTC diagnosis and classification is not quantified. What concerns me is that the authors have made it very clear in that response letter that the revised study is not reproducible and reliable. The authors even claimed statistics is not needed for the proposed study (which is a crucial reason why their study is not reliable). Therefore, this survey is not trustworthy and must be rejected.

Author Response

First and foremost, we sincerely apologize that our efforts to revise the manuscript did not satisfy your comments. It was never our intention to actively reject your suggestions. However, we hope you can understand that the main purpose of our manuscript is to “introduce” a new pathologic and molecular classification of biliary tract cancer (BTC), rather than to conduct a comprehensive bioinformatic analysis of this classification.

We acknowledge that there are indeed instances where the diagnosis of BTC presents uncertainties. Given the challenges in obtaining adequate cancerous tissue samples due to the anatomical location of BTC, particularly in early stages, diagnosis can be challenging. Typically, diagnosis of BTC is confirmed through surgical resection of suspicious lesions or clinical observation over a 6-months period to detect radiologic changes indicative of BTC. We believe these clinical scenarios have been adequately addressed in the sections of <5. Diagnostic tool > and <7. Approach to the patient>.

The uncertainty surrounding BTC diagnosis, as mentioned earlier, pertains to a minority of cases of BTC. The limited sample size and heterogeneity in diagnostic methods make replication and cross-validation with other large-sample studies challenging. Furthermore, there is a paucity of research in this area, contributing to the perceived limitations in “reproducibility and reliability”.

Most patients with BTC are diagnosed through pathologic examination. Approximately 40% of all patients with BTC underwent radical surgery, and the remainder were confirmed by tumor biopsy. For patients who have undergone surgery, both pathological and molecular classification is possible with surgical tissue. Whereas, for the remaining patients, pathological classification is usually difficult due to limited sample size. We introduced various molecular classifications of BTC through the newly incorporated <3. Molecular classification> section, but we recognize the absence of a universally accepted molecular classification, as well as the need for future research in this direction.

Regarding the perceived assertion that statistics are unnecessary, we believe there may have been a misunderstanding. Given that our target reader primarily comprises clinicians rather than bioinformatics experts, our manuscript aims to cater to their needs. We admit that conducting systematic surveys with statistical analyses exceeds our expertise. Should you deem it necessary for our manuscript to include such analyses for reproducibility and reliability, we defer to the editorial team's judgment on its acceptance.

Once again, we sincerely appreciate your thorough review and constructive feedback.

Reviewer 3 Report

Comments and Suggestions for Authors

The Authors have adequately addressed my concerns. Thank you.

Author Response

Once again, I would like to express my sincere gratitude for your work and dedication.